# *Elaeocarpus sylvestris* var. *ellipticus* Extract and Its Major Component, Geraniin, Inhibit Herpes Simplex Virus-1 Replication

**DOI:** 10.3390/plants13111437

**Published:** 2024-05-22

**Authors:** Yeong-Geun Lee, Dae Won Park, Jeong Eun Kwon, Hyunggun Kim, Se Chan Kang

**Affiliations:** 1Department of Oriental Medicine and Biotechnology, Kyung Hee University, Yongin 17104, Republic of Korea; lyg629@nate.com (Y.-G.L.); jjung@nmr.kr (J.E.K.); 2GENENCELL Co., Ltd., Yongin 16950, Republic of Korea; parkdw@genencell.co.kr; 3Department of Biomechatronic Engineering, Sungkyunkwan University, Suwon 16419, Republic of Korea

**Keywords:** *Elaeocarpus sylvestris*, herpes simplex virus–1, natural products inhibiting luciferase

## Abstract

*Elaeocarpus sylvestris* var. *ellipticus* (ES), which our research group had confirmed inhibits influenza A and SARS-CoV-2 viruses, was investigated to identify new potent and selective inhibitors of herpes simplex virus-1 (HSV–1) replication. To clarify the optimal condition for ES extract (ESE), ES was extracted at different concentrations of 0, 30, 50, 70, and 100%, to screen for its anti-HSV–1 effect. Among these ESE samples, ESE50 (50% concentration) exhibited the strongest inhibition of HSV–1 replication (EC_50_ 23.2 μg/mL) while showing low cytotoxicity on host cells (IC_50_ 342.8 μg/mL). The treatment of ESE50 clearly demonstrated a decrease in the expression of ICP0 in the lungs of HSV–1-infected BALB/c nude mice, compared to the MOCK group. Geraniin, which was isolated from ESE50 and analyzed using ESI−MS and 1D−(^1^H− and ^13^C−) and 2D−NMR, showed greater potency in inhibiting HSV–1 replication, as determined by the plaque reduction assay (EC_50_ 8.3 μg/mL) and luciferase inhibition (EC_50_ 36.9 μg/mL). The results demonstrate that ESE50 and geraniin show great potential as candidates for new drug discovery in the treatment of HSV–1 and related diseases.

## 1. Introduction

Herpes simplex virus type 1 (HSV–1) and type 2 (HSV–2) belong to the Herpesviridae family. They are enveloped large linear double-stranded DNA viruses that globally cause oral diseases, keratoconjunctivitis, and genital infections in humans [1,2,3]. In particular, HSV–1 is present in approximately 70% of the human population. While HSV–1 infection is generally mild, it can lead to various diseases, such as oral herpes, herpes simplex virus encephalitis, and herpes simplex keratitis [4,5,6]. HSV–1 has been widely used as a prototype to study productive and latent infections of herpesvirus [7]. During productive infection, HSV–1 efficiently redirects the host transcriptional machinery to express its own genes in a tightly regulated temporal cascade, consisting of the sequential expression of three gene classes: the immediate–early (IE), early (E), and late (L) genes. The five IE genes are expressed shortly after entry into the host cell, and the resultant IE proteins (infected-cell proteins (ICP)–0, –4, –22, –27, and –47) are essential for the subsequent temporally controlled expression of E genes that would encode proteins involved in viral DNA replication, and later the L genes that would predominantly encode structural proteins [8]. ICP0 is important in reactivation from latency [9,10]. The immediate–early protein ICP0 of HSV–1 plays a significant regulatory role throughout viral infection in cultured cells and in mouse latency models. While ICP0 is not essential for normal viral gene expression or replication, the inactivation of ICP0 results in cell type-, cell cycle-, and multiplicity-dependent defects in the onset of viral gene expression after productive infection has been initiated [11,12,13]. Therefore, in this study, we focused on examining whether the virus could be effectively suppressed at the ICP0 stage.

Anti-herpetic drugs, such as acyclovir (ACV) or penciclovir (PCV), and their structure- or function-related derivatives were developed as first-line drugs [14]. These drugs can be used to treat primary or recurrent infections. However, the long-term use of these anti-herpetics can result in the evolution of drug-resistant virus strains [15,16]. Therefore, there have been many attempts to develop new drugs with novel targets that can suppress viral replication and minimize the likelihood of drug resistance. In this respect, natural products have been a rich source of antiviral drugs [17]. *Elaeocarpus sylvestris* var. *ellipticus* (ES) is a species of tropical and subtropical evergreen trees and shrubs. Its region of distribution includes the subtropical zone, from Jeju island in South Korea to Okinawa and Kyushu in Japan, Southern China, and Taiwan. It has been reported that the ES extract (ESE) containing tannin (1,2,3,4,6–penta–*O*–galloyl–*β*–D–glucose, which is composed of gallates) inhibits the expression of inducible nitric oxide synthase and the production of prostaglandin E2 in RAW 264.7 cells [18]. Our previous study also demonstrated that the 70% ethanol extract of ES inhibited the replication of the human cytomegalovirus (HCMV) without significant adverse effects on the viability of human foreskin fibroblasts, and considerably downregulated the immediate–early gene expression of HCMV [19]. To the best of our knowledge, there has been no study that has reported the antiviral activity of ES for HSV–1. Geraniin, belonging to the tannins and composed of gallate and its derivates, was first isolated from the *Geranium thunbergii* Sieb. et Zucc in 1976, and reported to be the constituent in *Geranium* spp., Euphorbiaceae family, *Nephelium lappaceum*, and *Nymphaea tetragona* [20,21]. Geraniin is also isolated from the leaves of ES [22]. This compound has a large range of biological activities, and is known to have antioxidant [23], radioprotective [24], antihypertensive [25], antimicrobial [22], antitumor [26], and antinociceptive activities [27]. More importantly, geraniin has shown antiviral effects on human immunodeficiency virus [28], hepatitis B virus [29], and herpes simplex virus [30].

In the present study, we identify the active compound of ES, and investigate the antiviral activity of ES affecting HSV–1 infection and/or replication in vitro and in vivo. The inhibitory efficacy of the tested substances on virus infection and/or replication is evaluated by conducting luciferase activity and plaque reduction assays at the cellular level, as well as an in vivo assay using HSV–1 RNA quantification. The comparison is made against the control drug, ACV, to determine the potential of these agents as anti-HSV–1 treatments.

## 2. Results and Discussion

To clarify the optimization of ES extraction condition, ES was extracted at different concentrations of ethanol ((EtOH)/H_2_O) of 0/100, 30/70, 50/50, 70/30, and 100/0 *v*/*v* (ESE). The best extraction condition was found with the ES 50% ethanolic extract (ESE50), which demonstrated a better inhibition of plaque formation and an improved cell viability of Vero cells infected with HSV–1 (EC_50_ (23.2 ± 0.22) μg/mL, IC_50_ (342.8 ± 2.59) μg/mL; Table 1).

Based on these data, we selected ESE50 as the optimal extraction concentration for subsequent analysis. The immediate–early protein of HSV–1, ICP0, plays an important regulatory role during viral infection in cultured cells and in mouse latency models. According to cytotoxicity at various concentrations of the HSV-1 in vitro assay, the optimal concentration for HSV-1, namely, 5.75 × 10^7^ pfu per 200 μL, and the method for virus counting from the literature were employed [31,32]. Figure 1 shows the effect of the treatment of ESE on the expression of ICP0 in the lung of BALB/c nude mice infected with HSV–1. The ESE treatment significantly decreased the mRNA expression levels of ICP0 compared to the HSV–1-infected group (CON), at levels similar to the normal control group (MOCK) or the group with HSV–1 infection and 25 mg/kg bw/d (d) of acyclovir (ACV). These data indicate that ESE50 has potent antiviral activity leading to the inhibition of HSV–1 replication.

To characterize ESE50, a quantitative analysis of geraniin was performed by means of reversed-phase HPLC. The content of geraniin in ESE50 (18.03 min) was determined to be 132.16 μg/mg extract (Figure 2). In our previous study, it was demonstrated that the other major peak at 24.86 min referred to pentagalloyl glucose (PGG), with a quantity of 152.68 μg/mg [33]. The isolated compounds from ESE50 were analyzed using ESI−MS and ^1^H/^13^C NMR including 2D−NMR. The compounds were identified as ESE50 and geraniin by comparing their spectroscopic data with previously reported findings in the literature [33].

The antiviral evaluation of ESE50 and the geraniin of ES was performed using virus-cell-based assays for HSV–1 (Figure 3). ESE50 significantly reduced plaque formation and luciferase in the HSV–1-infected cells compared to the MOCK group, and did not affect cell viability. Also, the major compound, geraniin, markedly decreased plaque formation and luciferase in the same manner as ESE50. Together, these data suggest the important role of geraniin in inhibiting HSV–1 replication without affecting the viability of Vero cells. Therefore, it is inferred that geraniin could be the active compound of ESE50 with an antiviral effect against HSV–1.

As shown in Notka’s research, geraniin was demonstrated to effectively inhibit HSV-1 replication. According to the research findings, geraniin inhibits the activity of HSV-1 reverse transcriptase without showing cytotoxicity, thereby effectively controlling HSV-1 infection [28]. These data also correspond to the previous study that indicated that geraniin inhibits HSV–1 in vitro in this research.

ESE50 and geraniin demonstrate inhibitory effects against the HSV-1 virus, suggesting potential applications as antiviral therapeutics. Furthermore, their antiviral activity opens up new possibilities for the development of novel antiviral agents.

Our evaluation is currently limited to in vitro and in vivo models, necessitating further research to confirm the efficacy of ESE50 in clinical trials. Additionally, investigations into its precise mechanisms of action and interactions with diverse viruses are imperative. Such endeavors are pivotal for advancing our understanding and paving the way for the development of more efficacious antiviral therapies.

## 3. Materials and Methods

### 3.1. Analytical Procedures for Phytochemical Analysis

Analytical thin-layer chromatography (TLC) was performed on precoated silica gel F_254_ plates (Spectra/Chrom). High-performance liquid chromatography (HPLC) was performed using a Knauer Smartline System, Berlin, Germany (Manager 5000, UV detector 2500). The ^1^H and ^13^C NMR spectra were recorded at room temperature using Bruker Avance 600 (Billerica, MA, USA) NMR spectrometer, while HR−ESI/MS was measured on a Triple TOF 5600^+^ (AB Sciex, Framingham, MA, USA).

### 3.2. Sample Preparation

*Elaeocarpus sylvestris* var. *ellipticus* (Specimen number JBR-083) was harvested and collected in Jeju Island, Republic of Korea, and deposited at the BMRI, Kyung Hee University (Yongin, Republic of Korea). The preparation of sample, isolation procedure, and chemical characterization have previously been described [31]. The ^1^H NMR spectra of the samples were recorded with a 600 MHz a Bruker Avance 600 (Billerica, MA, USA), using tetramethylsilane as an internal reference, while the ^13^C NMR spectra were run at 150 MHz (Appendix A).

### 3.3. High-Performance Liquid Chromatography Analysis

HPLC was performed with a Knauer Smartline System (Manager 5000, UV detector 2500) equipped with a Phenomenex Gemini NX 5 μm C18 110A column (150 mm × 4.6 mm). To characterize ESE50, a quantitative analysis of geraniin was performed by reversed-phase C−18 HPLC analysis with gradient elution. The eluent consisted of 0.02% formic acid in MeOH (A), and 0.02% formic acid in water (B). The gradient profile was as follows: 0–6 min, isocratic 5% A in B; 6–10 min, linear change from 5 to 15% A in B; 10–20 min, linear change from 15 to 20% A in B; 20–30 min, linear change from 20 to 30% A in B; 30–35 min, linear change from 30 to 70% A in B; and 35–45 min, isocratic 70% A in B. The UV absorption was measured at 254 nm, and the flow rate and column oven temperature were set at 1 mL/min and 30 ℃, respectively. The content of geraniin in ESE50 was determined using the standard curve.

### 3.4. Cytotoxicity Assay

Vero cells were plated, treated with ESE50 and geraniin, and incubated as described above in the tetrazolium-dye (MTT) assay. For the MTT cytotoxicity assays, 10 μL of the MTT (Sigma, St. Louis, MO, USA, 5 mg/mL) solution was added to each well. The plates were incubated for 3 h at 37 °C, the supernatant was removed from the wells, and 100 μL of DMSO was added to the wells to dissolve the MTT crystals. The plates were then placed on a shaker for 10 min, and the absorbency was read at 492 nm on a multi-reader (Tecan, Männedorf, Switzerland).

### 3.5. Antiviral Assays

The antiviral activities of ESE50 and geraniin were determined using the HSV–1 plaque reduction assays and luciferase assays. HSV–1 (American Type Culture Collection, Rockville, MD, USA) was propagated in African green monkey kidney cells (Vero cells; Korean Cell Line Bank, Seoul, Republic of Korea, KCLB 10081), The Vero cells were in RPMI−1640 (GIBCO−BRL, Gaithersburg, MD, USA) supplemented with 10% fetal bovine serum (Hyclone Laboratories, Logan, UT, USA), 100 units/mL penicillin, 100 μg/mL streptomycin, and 25 mM of 4–(2–hydroxyethyl)–1–piperazineethanesulfonic acid (HEPES). The procedures used for the antiviral assays have been previously described [19]. The plaque reduction assays were performed with confluent monolayer Vero cells and 0.3 m.o.i. of HSV–1. Virus was adsorbed to cells in RPMI−1640 without serum for 2.5 h at 37 ℃, and then removed and overlaid with agarose. Each dilution of ESE50 and its active compound was plated in duplicate. After 4 d, the plaques in each well were counted, and then expressed as a percentage of the control at each concentration. For the cells transfected with RLuc-expressing plasmids, the cells were lysed in a passive lysis buffer, and RLuc and luciferase activities were assayed using a luciferase assay system (Promega, Madison, WI, USA) and a GlpMax 20/20 single-tube luminometer (Carlsbad, CA, USA).

### 3.6. Animals

The experimental protocol (KHUAGC-17-002) was reviewed and approved by the Animal Experiment Committee of Kyung Hee University (Yongin, Republic of Korea). All animals were maintained and managed in accordance with the Kyung Hee University Animal Use and Care protocols. All experimental procedures were undertaken in compliance with the Guide for the Care and Use of Laboratory Animals (National Institutes of Health, Bethesda, MD, USA) and the National Animal Welfare Law of the Republic of Korea. Four-week-old male BALB/c nude mice were obtained from SLC Inc. (Shizuoka, Japan), and maintained in a controlled environment of 22 ± 1 °C and a humidity of 50 ± 10% with a 12 h light–dark cycle for 1 week, prior to the commencement of the experiments. The mice were allowed access to sterile standard mouse chow and water ad libitum.

### 3.7. In Vivo Conditions

The BALB/c nude mice were given 5.75 × 10^7^ pfu per 200 μL of virus by the intraperitoneal route. The mice were randomly divided into 6 groups (n = 5 each) as follows: (ⅰ) normal control group (PBS; MOCK); (ⅱ) HSV–1 infection control (PBS; CON); (ⅲ) HSV–1 infection and 25 mg/kg bw/d of ESE50; (ⅳ) HSV–1 infection and 50 mg/kg bw/d of ESE50; (ⅴ) HSV–1 infection and 100 mg/kg bw/d of ESE50 via oral gavage for 10 d; and (ⅵ) HSV–1 infection and 25 mg/kg bw/d of acyclovir (ACV) via intraperitoneal for every 2 d. The mice were sacrificed after 30 d, and RNA was isolated from the lungs and stored at −80 °C.

### 3.8. In Vivo Protocols

The samples for extracting RNA were obtained from the lung tissue by means of cryogenic grinding, and total RNA was extracted using the PureLink^™^ RNA Mini Kit (Ambion, Austin, TX, USA). One μg of total RNA in 20 μL volume was transcribed using oligo (dT) primers, with the enzyme and buffer supplied in the PrimeScript Ⅱ 1st strand cDNA Synthesis kit (Takara, Osaka, Japan). Quantitative real-time PCR (qRT−PCR) was performed using a MX3005P (Stratagene, La Jolla, CA, USA) with the following primers: ICP0, forward 5′-AAGCTTGGATCCGAGCCCCGCCC-3′ and reverse 5′-AAGCGGTGCATGCACGGGAAGGT-3′; *β*-actin, forward 5′-ATCATGTTTGAGACCTTCAAC-3′ and reverse 5′-CAGGAAGGAAGGCTGGAAGAG-3′. The relative quantitative evaluation of ICP0 was performed using a comparative cycle threshold (CT).

## 4. Conclusions

In the present study, we provide chemical and biological evidence supporting the effectiveness of a potent antiviral agent, ESE50, along with its active compound, geraniin. Our findings demonstrate that ESE50 effectively inhibits plaque formation and the luciferase activity of HSV–1. Notably, geraniin, which is a major component of ESE50, plays a key role in antiviral activity against HSV–1. These discoveries position ESE and geraniin as promising candidates for potential drug development in the treatment of HSV–1 and related diseases. However, further research is required to ascertain the underlying mechanism of ESE in reducing HSV–1 infection.

## Figures and Tables

**Figure 1 plants-13-01437-f001:**
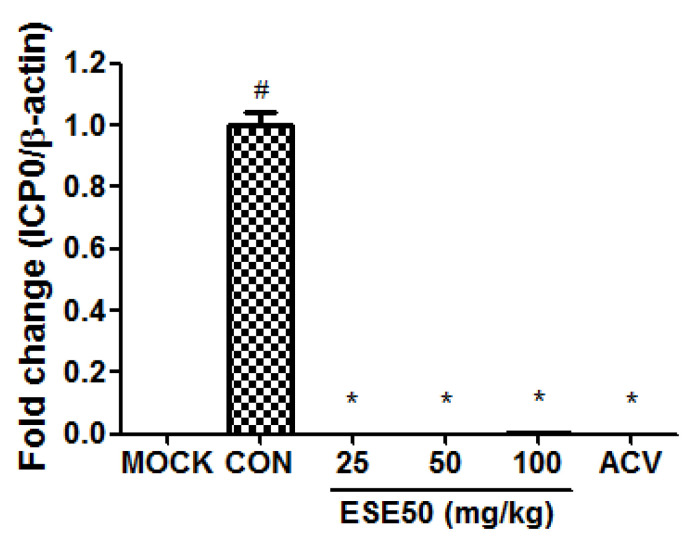
Anti-HSV–1 effect of treatment with ESE50 in the HSV–1-infected mouse model. ^#^ Different from the MOCK group (*p* < 0.05); * Different from the untreated CON group (*p* < 0.05). MOCK, normal control group; CON, HSV–1 infection control; ACV, HSV–1 infection and 25 mg/kg bw/d of acyclovir.

**Figure 2 plants-13-01437-f002:**
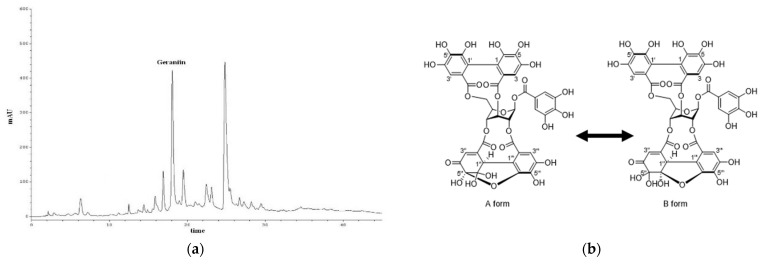
(**a**) HPLC chromatograms of ESE50, and (**b**) chemical structures of the isolated components.

**Figure 3 plants-13-01437-f003:**
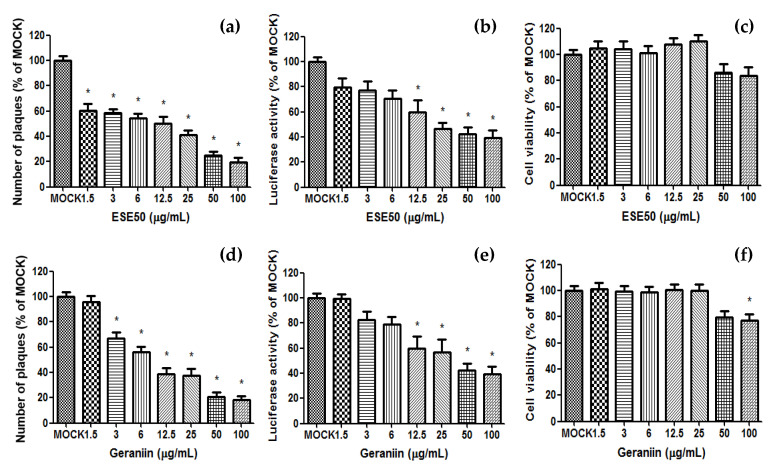
Antiviral activity of ESE50 and its components against HSV–1 in Vero cells. Vero cells cultured in a six-well tissue culture plate were infected with HSV-1 (HF strain) at 0.3 m.o.i./well for 2.5 h, and then removed and overlaid with agarose. They were cultured for 4 days with DMEM containing diluted ESE50 and geraniin. (**a**,**d**) Cell sheets were stained and fixed with crystal violet, and the total number of plaques was quantified. (**b**,**e**) For luciferase activity, RLuc and luciferase activities were assayed using a luciferase assay system and a GlpMax 20/20 single-tube luminometer. (**c**,**f**) Cell viability was measured using the MTT assay. * Different from the MOCK group (*p* < 0.05). MOCK, normal control group; CON, HSV–1 infection control.

**Table 1 plants-13-01437-t001:** Cytotoxicity and antiviral activity of the ES extracts at different EtOH concentrations.

EtOH (%)	0	30	50	70	100
EC_50_ ^1^	143.8 ± 2.51	42.6 ± 0.85	23.2 ± 0.22	28.5 ± 0.37	63.8 ± 1.82
IC_50_ ^2^	>500	358.5 ± 2.37	342.8 ± 2.59	209.4 ± 3.61	184.3 ± 1.68

^1^ Data in μg/mL. Antiviral activity was determined by means of the plaque reduction assay on HSV–1. ^2^ Data in μg/mL. Cytotoxicity was assessed with the MTT assay on Vero cells. All data are expressed as the mean ± SD from three independent experiments.

## Data Availability

No new data were created or analyzed in this study. Data sharing is not applicable to this article.

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
