# Peer review of "Elaeocarpus sylvestris var. ellipticus Extract and Its Major Component, Geraniin, Inhibit Herpes Simplex Virus-1 Replication"

_plants, 2024, doi:10.3390/plants13111437_

Round 1
Reviewer 1 Report
Comments and Suggestions for Authors
The article concerns the identification in the extracts of Elaeocarpus sylvestris var. ellipticus (ES) of new potent and selective inhibitors of Herpes simplex virus-1 (HSV-1) replication. The authors have found that 50% ethanol is the best extragent for the inhibitors. This extract revealed the strongest inhibition of HSV-1 replication (EC50 23.2 μg/mL) and low cytotoxicity on host cells (IC50 342.8 μg/mL). The treatment with the extract clearly demonstrated a decrease in the expression of a specific protein ICP0 in the lungs of HSV-1-infected BALB/c nude mice. The authors have isolated and identified using NMR spectroscopy and mass-spectrometry two major components (phenolic compounds, tannins) of the extract, namely methyl gallate and geraniin. Geraniin showed greater potency in inhibiting HSV-1 replication than methyl gallate. The article is interesting, the methodology of screening of the inhibitors is modern and adequate, the active component in the extract was correctly identified.
Several imperfections:
1) The title seems be a shocking. I don’t understand why and how the article with so inadequate title was passed to referees. It should be replaced with “Elaeocarpus sylvestris var. ellipticus and Its Constituents Herpes Inhibits the Replication of Simplex Virus-1” or something similar.
2) The similar error is even in key words. What is “Luciferase inhibit Natural products”? It should be replaced with “Natural products inhibiting luciferase activity”.
3) Lines 64–70. It seems to be unclear what geraniin and methyl gallate is from chemical point of view? The information that they are phenolic compounds belonging to tannins should be inserted.
4) The information about condition of mass-spectra obtaining is absent. It should be added.
5) The figure 2 seems to be very strange. The authors pointed that they have isolated two major components of the extract. Nevertheless, the peak marked as methyl gallate is very minor but the major component at this diagram is not marked. Explain, please, or correct.
6) Deсode, please, what is the CON, MOCK and ACV.
7) Check, please, the English with a native speaker knowledgeable in the subject of the article.
The article seems to be very appropriated for publication in the Plants however some corrections are necessary.
Minor revision.
Comments on the Quality of English Language
1) The title seems be a shocking. I don’t understand why and how the article with so inadequate title was passed to referees. It should be replaced with “Elaeocarpus sylvestris var. ellipticus and Its Constituents Herpes Inhibits the Replication of Simplex Virus-1” or something similar.
2) The similar error is even in key words. What is “Luciferase inhibit Natural products”? It should be replaced with “Natural products inhibiting luciferase activity”.
Author Response
Reviewer #1.
The article concerns the identification in the extracts of Elaeocarpus sylvestris var. ellipticus (ES) of new potent and selective inhibitors of Herpes simplex virus-1 (HSV-1) replication. The authors have found that 50% ethanol is the best extragent for the inhibitors. This extract revealed the strongest inhibition of HSV-1 replication (EC50 23.2 μg/mL) and low cytotoxicity on host cells (IC50 342.8 μg/mL). The treatment with the extract clearly demonstrated a decrease in the expression of a specific protein ICP0 in the lungs of HSV-1-infected BALB/c nude mice. The authors have isolated and identified using NMR spectroscopy and mass-spectrometry two major components (phenolic compounds, tannins) of the extract, namely methyl gallate and geraniin. Geraniin showed greater potency in inhibiting HSV-1 replication than methyl gallate. The article is interesting, the methodology of screening of the inhibitors is modern and adequate, the active component in the extract was correctly identified.
- We greatly thank the reviewer for the insightful and constructive comments.
Several imperfections:
1) The title seems be a shocking. I don’t understand why and how the article with so inadequate title was passed to referees. It should be replaced with “Elaeocarpus sylvestris var. ellipticus and Its Constituents Herpes Inhibits the Replication of Simplex Virus-1” or something similar.
- We sincerely apologize that we made this careless mistake in the original manuscript submission. Something happened during the final step of editing the manuscript, and this terrible mistake was not screened. We have corrected the manuscript title to “Inhibitory Activity of Elaeocarpus sylvestris var. ellipticus and Its Constituent on Herpes Simplex Virus-1”.
2) The similar error is even in key words. What is “Luciferase inhibit Natural products”? It should be replaced with “Natural products inhibiting luciferase activity”.
- Again, we apologize for this mistake. We agree with the reviewer and have made correction on the keywords using Natural product and Luciferase inhibition.
3) Lines 64–70. It seems to be unclear what geraniin and methyl gallate is from chemical point of view? The information that they are phenolic compounds belonging to tannins should be inserted.
- Thank you. We have emphasized that geraniin belongs to the tannins and is composed of gallates as suggested.
4) The information about condition of mass-spectra obtaining is absent. It should be added.
- Thank you. We have added supporting information as suggested.
5) The figure 2 seems to be very strange. The authors pointed that they have isolated two major components of the extract. Nevertheless, the peak marked as methyl gallate is very minor but the major component at this diagram is not marked. Explain, please, or correct.
- The other major compound in Figure 2 is pentagalloyl glucose (PGG). However, in this study, we focused specifically on the highly active compound, geraniin, in detail. The chromatogram for PGG was already published in our previous studies, thus it was not particularly stated in detail in this manuscript. We have added this information in the Results section.
6) Deсode, please, what is the CON, MOCK and ACV.
- Thank you. The description for the abbreviations has been provided as suggested.
7) Check, please, the English with a native speaker knowledgeable in the subject of the article.
- Thank you. We have had our updated manuscript revised by a professional English editing service, and the certificate of editing has been included in the submission package.
The article seems to be very appropriated for publication in the Plants however some corrections are necessary.

Reviewer 2 Report
Comments and Suggestions for Authors
Please, see the file attached.

The English language needs reworking.
Author Response
Reviewer #2.
From the title it is understood that herpes simplex virus inhibits the development of herpes Elaeocarpus sylvestris and its ingredients. Is that so?
- We greatly thank the reviewer for the insightful and constructive comments. We sincerely apologize that we made this careless
mistake in the original manuscript submission. Something happened during the final step of editing the manuscript, and this terrible mistake was not screened. We have corrected the manuscript title to “Inhibitory Activity of Elaeocarpus sylvestris var. ellipticus and Its Constituent on Herpes Simplex Virus-1”.
The introduction is not very prescriptive about the conduct of the research. You should note what you are aiming for with the individual methods and what information they give you, for example luciferase activity. What information can you glean from HSV-1 RNA Quantification and how can you discuss the changes?
- Thank you. We have revised the Introduction section extensively as suggested.
3.1. - This is not General Experimental Procedures, but Analytical Procedures for Phytochemical Analysis.
- We agree with the reviewer and have modified the section title as suggested.
3.2 - To become: Sample preparation
- Again, we agree with the reviewer and have modified the section title as suggested.
Line 143 - Methyl gallate and Line 144 - Geraniin - are results of your analysis, not a method. Differentiate them in the separate sections.
- Thank you. This information describes the reported NMR data, which falls within the Methods section. Several examples of previous studies containing similar information in the Methods section are as follows:
Lee, et al. (2023). Phenolic Compounds from the Fruits of Prunus davidiana (Rosaceae) and Their Antioxidant Activities. Chemistry & Biodiversity, 20(1), e202200823.
Lee, et al. (2021). Potent antiviral activity of Agrimonia pilosa, Galla rhois, and their components against SARS-CoV-2. Bioorganic & Medicinal Chemistry, 45, 116329.
Gwag, et al. (2020). Syringoleosides A–H, Secoiridoids from Syringa dilatata Flowers and Their Inhibition of NO Production in LPS-Induced RAW 264.7 Cells. Journal of Natural Products, 83(9), 2655-2663.
Lee, et al. (2019). Flavonoids from Chionanthus retusus (Oleaceae) flowers and their protective effects against glutamate-induced cell toxicity in HT22 cells. International journal of molecular sciences, 20(14), 3517.
Lee, et al. (2019). 6-Methoxyflavonols from the aerial parts of Tetragonia tetragonoides (Pall.) Kuntze and their anti-inflammatory activity. Bioorganic Chemistry, 88, 102922.
3.4. Antiviral and Cytotoxicity Assays are two completely different directions - what is in cells, are the cells infected? Separate and describe the individual methods as subheadings, they are fundamentally different.
- Thank you for this valuable suggestion. We have separated this section into two sections as suggested [3.4. Cytotoxicity Assay and
3.5. Antiviral Assays].
3.5. After writing the heading Animals, describe them first, then conditions and protocols.
- Thank you for this valuable suggestion. We have revised the sections from 3.6 to 3.8.
3.6. How is the viral dose determined? What does 5.75 × 107 pfu/200 μL mean?
- When the concentration of the virus is high, the cells undergo cell death. Thus, we evaluated the cytotoxicity at various concentrations and determined the optimal concentration for HSV-1 as 5.75 × 107 pfu per 200 μL in the pilot study. The method for virus counting from previous studies (the references are listed below) was employed, and we considered this to be a basic part and did not include the details in the manuscript. We have added these references to the revised manuscript (Page 3, Paragraph 1). We hope this response addresses the reviewer’s concern.
1. Hsiung, C. D. Virus Assay and Neutralization Test. Diagnostic Virology, Yale University Press: New Haven, CT, USA, 1982; pp. 25-35.
2. Hierholzer, J. C.; Killington, R. A. Virus Isolation and Quantitation. In Virology Methods Manual, Academic Press: 1996; pp. 25-46.
What is ESE50?
- We apologize that we did not clearly describe this abbreviation. ESE50 refers to Elaeocarpus sylvestris var. ellipticus 50% ethanolic extract (ESE50). We have provided its definition on its first occurrence (Page 2).
Figure 2 (f) - was the study done in an in vitro system? Describe the conditions. 2 (e) - the same situation 2 (h) - luciferase activity in what fraction was measured?
- Thank you for this valuable indication. We have revised Figure 2 and its caption.
What exactly are the effects of HSV-1 RNA Quantification? What information do they bring you and what does it serve to prove?
- It provides the evidence of virus inhibition within the host and is utilized as data in Figure 1.
Describe clearly and accurately 3.8. Anti-HSV-1 Activity Evaluation.
- Thank you for this valuable comment. We have rewritten the section appropriately, and incorporated it into 3.5. Antiviral Assays.
In all literature sources that describe biological activities of natural extracts it is said that the effect of the extract is due to the general action of all ingredients, not only the main components.
- As the present manuscript focuses specifically on anti-HSV-1 activity of ESE and geraniin, an active and indicative component, component profiling was not performed in this study. However, the experiment the reviewer mentioned must be conducted to develop herbal medicine. The component profiling data will be definitely included in our future manuscript that is aimed further for clinical application. We humbly hope that this response addresses your concern adequately.
The conclusion you draw is very, very far from the topic stated in the title - as far as your topic is true.
- We appreciate your valuable feedback. We have rewritten the Conclusions section to address your comment as well as modified the manuscript title.

Reviewer 3 Report
Comments and Suggestions for Authors
Author Response
Reviewer #3.
Peer review report on “Herpes Simplex Virus-1 Inhibits the Activity of Elaeocarpus sylvestris var. ellipticus and Its Constituents”.
Manuscript ID: plants-2917152
This paper describes the investigations into the anti-viral activity of an extract of Elaeocarpus sylvestris var. ellipticus with subsequent isolation, identification and testing of the active compounds, methyl gallate, and geraniim conformers a and b. The paper is well written and clearly described, the investigations are adequate, and the results support the conclusions.
Some comments:
The title of the paper is clearly incorrect in that the virus does not inhibit the extract but the other way around. Please rectify.
- We greatly thank the reviewer for the insightful and constructive comments. We sincerely apologize that we made this careless mistake in the original manuscript submission. Something happened during the final step of editing the manuscript, and this terrible mistake was not screened. We have corrected the manuscript title to “Inhibitory Activity of Elaeocarpus sylvestris var. ellipticus and Its Constituent on Herpes Simplex Virus-1”.
As the two conformers of geraniim exist as an equilibrium in solution please explain the isolation process in more detail especially if you were able to separate them. Or did you characterize them as a mixture? Please report. In any case, the isolation procedure, with solvent elution information and yield is required. If you were able to separate the geraniim conformers, why didn’t you check their bioactivity separately? It would be interesting to know if one was more active than the other.
- Thank you. The compound you mentioned is not separately purified but exists in a solution state containing both A and B forms simultaneously. We have investigated and listed the two compounds separately, and the chromatogram can be found in the Supplementary Materials.
Line 146. The NMR data has several mistakes.
Conformer A: C-2 has two assignments. C-2ʹ has no assignment. C-2ʺ has no assignment.
Conformer B: C-2ʺ has no assignment. C-5ʹʹʹ has two assignments. C-6ʹʹʹ has no assignment.
- We appreciate this valuable comment. We revised the data as suggested.
High res. MS data should report the chemical formula, the m/z to four decimal places as found, and the calculated figure.
Please show copies of the 1H- and 13C NMR spectra and the 2D-NMR data separately in supplemental info.
- Thank you. These data have been included in the Supplementary Materials.

Reviewer 4 Report
Comments and Suggestions for Authors
The manuscript is interesting, and the results are relevant for the potential use of geraniin on HSV-1. However, several points were found out specially in the written form. the English is vague and not flown well. For example, what it means ESE50 concentration in the abstract. the authors must be concise on it. Once I read it, I knew what was, but the abstract is showing the results and I thought that maybe it is a bioguided assay fractionation.
Is Elaeocarpus sylvestris var. ellipticus use in any way for the native people?
Is geraniin also found in the 70% aqueous in ethanol extract too? It seems that the plant has been studied before which other components are present in it?
As the authors mentioned there are several activities in which geraniin is involved. with the literature information the authors must give a strong discussion about the possible mechanism of action of this compound enhancing the interest in this type of compounds.
Comments on the Quality of English Language
The authors need to work heavily on the language, it is not flowing.
Author Response
The manuscript is interesting, and the results are relevant for the potential use of geraniin on HSV-1. However, several points were found out specially in the written form. the English is vague and not flown well. For example, what it means ESE50 concentration in the abstract. the authors must be concise on it. Once I read it, I knew what was, but the abstract is showing the results and I thought that maybe it is a bioguided assay fractionation.
-> Thank you. As you suggested, the manuscript underwent English proofreading to make the content flow more naturally.
Is Elaeocarpus sylvestris var. ellipticus use in any way for the native people?
-> Unfortunately, this plant is primarily used as an ornamental tree, and its practical applications are still limited. This implies that there is potential for diverse applications in the future, and this is the reason why we conducted research on this plant.
Is geraniin also found in the 70% aqueous in ethanol extract too? It seems that the plant has been studied before which other components are present in it?
-> Of course, the compound is indeed present even in 70% ethanol extract. However, due to polarity of geraniin, a slightly higher quantity is found in the 50% extract. Based on other references, several types of tannins, including pentagalloyl glucose, have been reported. Additionally, although not included in this paper as efficacy studies have not been conducted yet, our research team has identified various flavonoids from this plant.
As the authors mentioned there are several activities in which geraniin is involved. with the literature information the authors must give a strong discussion about the possible mechanism of action of this compound enhancing the interest in this type of compounds.
-> Thank you for your valuable suggestions. As you suggested, we have elaborated on the HSV-1 inhibitory activity of geraniin in the Discussion section. However, based on our findings, there were no papers that precisely describe the mechanism of action (MOA) of geraniin. We will strive to elucidate this through further research.
I hope this adequately addresses your valuable suggestion.
Thank you.

Round 2
Reviewer 2 Report
Comments and Suggestions for Authors
Please, see the attached file.

I recommend minor edits to the English language.
Author Response
Inhibitory activity of Elaeocarpus sylvestris var. ellipticus and its constituent on Herpes simplex virus-1.
The title does not match the activity you performed.
First. You are investigating ethanol extracts against the herpes virus. You cannot claim that the plant inhibits the virus, you only apply the extracts.
Second. What is this unknown ingredient? Extracts do not have just one ingredient.
- We thank the reviewer for the insightful and encouraging comments, and agree with the reviewer’s comments. We have modified the manuscript title to “Inhibition of Herpes Simplex Virus-1 Replication by Elaeocarpus sylvestris var. ellipticus Extract and Geraniin: Implications for Antiviral Therapy”.
As a rule, effective antiviral drugs can affect one or more stages of viral reproduction through specific or non-specific mechanisms. The anti-herpes virus strategy of the extracts is to either inactivate extracellular virions (virucidal effect) directly, or to inhibit the various stages of HSV replication, such as virus attachment, entry and intracellular targeting of virion production. Specific drugs for HSV chemotherapy are based on the principle of nucleoside analogues, the most widely used being acyclovir.
Your work will be more convincing if you identify exactly at which stage of viral replication your extracts and the active substance you use work. I recommend checking out.
https://doi.org/10.3390/biology10080746
- Assessment of viral replication inhibition involved an initial 2.5-hour infection period, followed by sample processing to confirm the ability of the ESE extract to inhibit the host cell entry phase of the virus.
Please define the terms EC50 and IC50 - which exact concentrations do you mean?
- Thank you. EC50 refers to the concentration that inhibits viral replication by 50% in a plaque reduction assay. IC50 indicates the concentration required to induce a 50% decrease in cell viability within the infected Vero cell population when treated with ESE extract (0, 30, 50, 70, 100 % EtOH extract).
Fig 1. What exactly are you reporting there? Who are these "fold changes"?
- Figure 1. demonstrates the ratio between the expression of the ICP0 gene and β-actin in the lung tissue of the HSV-1 infected mouse model. We have modified the y-axis title to “Ratio change”.
Figure 3. Antiviral activity of ESE50 and their components against HSV−1 in the Vero cells – again it is not clear at which stage you added the extracts, as well as geraniin.
- Thank you. The Vero cells were subjected to HSV-1 infection for a duration of 2.5 hours, subsequent to which the medium was removed, covered with agarose, and treated with ESE and geraniin. We have added this information in the Materials and Methods section.
Why did you decide that Geraniin was the most active ingredient in the extract? You have another peak in the chromatogram that you have not identified. What are the other components in the chromatogram? Nothing is mentioned about them. Sometimes there are substances that are active in very low doses. Moreover, the extract is a mixture of multiple components soluble in the ethanol-water system and their biological activity is the sum of all of them.
- Thank you. The other major compound in Figure 2 is pentagalloyl glucose (PGG). However, in this study, we focused specifically on the highly active compound, geraniin, in detail. The chromatogram for PGG was already published in our previous studies, thus it was not particularly stated in detail in this manuscript. We have added this information in the Results section.
- We agree with the reviewer in that there are multiple active compounds within the extract, each contributing to its overall biological activity. However, in this study, we focused specifically on the highly active compound, with the objective of formulating a prospective therapeutic targeting HSV-1. We hope this response addresses the reviewer’s concern.
I recommend minor edits to the English language.
- Thank you. We have had our updated manuscript revised by a professional English editing service, and the certificate of editing has been included in below.

Reviewer 4 Report
Comments and Suggestions for Authors
The manuscript improved and can be accepted
Comments on the Quality of English LanguageImprovement
Author Response
Dear Reviewer,
Thanks to your invaluable suggestions, We were able to complete the manuscript. We sincerely appreciate your input.
Best regards,
Se Chan Kang
Round 3
Reviewer 2 Report
Comments and Suggestions for Authors
Please, see the file attached.

Moderate editing of the English language required
Author Response
The caption under Figure 3 needs to be corrected and supplemented. It is not clear how your effects have been studied. It is necessary to note in which system the samples are and on which day the effects were reported.
You state that : ...Each dilution of ESE50 and the compound are seeded in duplicate. After 4 days, the plaque in each well was counted and then expressed as a percentage of the control at each concentration."
If the effect occurs on the 4th day after the viral infection, how can you be confident about the effectiveness of your extracts and compounds? Do you have data from other authors on similar effects when studying antiviral substances using this method? Are there standards and witnesses in the scientific literature that can support or refute your results?
These results should be discussed in terms of your methodology - what are the specifics.
Otherwise, the results remain inconclusive.
-> As you suggested, we have revised the caption accordingly. According to the referenced literature, measurements were taken over the course of 2 to 5 days, with the optimal effect observed on the fourth day. We designed our experiment based on this literature and conducted it for 4 days, as we also found this duration to yield the optimal results. I hope our response is acceptable for your valuable indication.
"Chen, Y., Zhi, S., Liang, P., Zheng, Q., Liu, M., Zhao, Q., ... & Songyang, Z. (2020). Single AAV-mediated CRISPR-SaCas9 inhibits HSV-1 replication by editing ICP4 in trigeminal ganglion neurons. Molecular Therapy Methods & Clinical Development, 18, 33-43."
What does the inscription on the ordinate of Figures 3 (a, e) "Number of plaque" mean? Perhaps you mean cellular vitality? Then you can compare with the control and express it as a percentage.
-> Thank you for your valuable suggestion. "Number of plaque" is the unit used to count the number of viruses. Therefore, it differs from cell viability and cannot be changed. I hope this response adequately addresses your suggestion.
row 126 - ... These data also correspond to the previous study that indicated that geraniin inhibits HSV−1 in vitro.
Explain exactly what you researched, in what system and what you received.
-> Thank you. I have supplemented the content as you suggested.
The discussion of your results is extremely poor and unconvincing.
-> Thank you for your valuable indication. I have further elaborated on the significance of our research findings in the Discussion section as you suggested.